# Development of Aloe Vera-Green Banana Saba-Curcumin Composite Film for Colorimetric Detection of Ferrum (II)

**DOI:** 10.3390/polym14122353

**Published:** 2022-06-10

**Authors:** Joseph Merillyn Vonnie, Bong Jing Ting, Kobun Rovina, Kana Husna Erna, Wen Xia Ling Felicia, Nasir Md Nur ‘Aqilah, Roswanira Abdul Wahab

**Affiliations:** 1Faculty of Food Science and Nutrition, Universiti Malaysia Sabah, Jalan UMS, Kota Kinabalu 88400, Sabah, Malaysia; vonnie.merillyn@gmail.com (J.M.V.); bongjingting@gmail.com (B.J.T.); mn1911017t@student.ums.edu.my (K.H.E.); felicialingling.97@gmail.com (W.X.L.F.); aqilah98nash@gmail.com (N.M.N.‘A.); 2Department of Chemistry, Faculty of Science, Universiti Teknologi Malaysia, Johor Bahru 81310, Johor, Malaysia; roswani@utm.my

**Keywords:** bio-film, heavy metals, natural source, sensor

## Abstract

This study was performed to develop and characterize a bio-film composed of Aloe vera (*Aloe barbadensis*), green banana Saba (*Musa acuminata* x *balbisiana*), and curcumin for the detection of Fe^2+^ ions. Cross-linking interaction between banana starch-aloe vera gel and banana starch-curcumin enhanced l the sensing performance of the composite film towards divalent metal ions of Fe^2+^. The morphological structure of the Aloe vera-banana starch-curcumin composite revealed a smooth and compact surface without cracks and some heterogeneity when observed under Scanning Electron Microscopy (SEM). The thickness, density, color property, opacity, biodegradation, moisture content, water-solubility, water absorption, swelling degree, and water vapor permeability of bio-films were measured. The incorporation of aloe vera gel and curcumin particles onto the banana starch film has successfully improved the film properties. The formation of the curcumin-ferrum (II) complex has triggered the film to transform color from yellow to greenish-brown after interaction with Fe^2+^ ions that exhibit an accuracy of 101.11% within a swift reaction time. Good linearity (R^2^ = 0.9845) of response on colorimetric analysis was also obtained in Fe^2+^ ions concentration that ranges from 0 to 100 ppm, with a limit of detection and quantification found at 27.84 ppm and 92.81 ppm, respectively. In this context, the film was highly selective towards Fe^2+^ ions because no changes of color occur through naked eye observation when films interact with other metal ions, including Fe^3+^, Pb^2+^, Ni^2+^, Cd^2+^, and Cu^2+.^ Thus, these findings encourage curcumin-based starch films as sensing materials to detect Fe^2+^ ions in the field of food and agriculture.

## 1. Introduction

Heavy metal ions are hazardous compounds that threaten living things and the environment. Heavy metals are non-biodegradable, water-soluble, have a long half-life, and accumulate in food chains and living organisms. Heavy metal contamination is mainly due to the chemicals generated from industrial and mining activity, household wastes, fossil fuel burning, fertilizers, cosmetics, and their by-products [1]. Heavy metals can enter the human body through ingestion, inhalation, and skin contact [2]. When heavy metals are consumed or inhaled, they bio-accumulate in our body system [3]. Among various heavy metals, lead (Pb), cadmium (Cd), mercury (Hg), chromium (Cr), and arsenic (As) are highly toxic as they might lead to severe problems for the environment and human health even at small doses. Nonetheless, some essential metals such as iron (Fe), zinc (Zn), copper (Cu), cobalt (Co), and manganese (Mn) only become hazardous when present in high doses [4]. For instance, iron pollution might develop as a result of pipe corrosion, water supply from groundwater systems, and rainwater. Groundwater and rainwater are prominently considered to be an important supply of drinking water in rural areas, which have a probability of causing iron toxicity [5,6]. Iron toxicity is a prevalent type of poisoning caused by intentional or unintentional ingestion, which might cause vomiting, diarrhea, nausea, abdominal pain, dehydration, and fatigue [7]. 

Curcumin, known as diferuloylmethane, is the primary natural polyphenol found in the rhizome of *Curcuma longa* (turmeric) and other *Curcuma* spp. [8]. As curcumin exhibits optical properties as a fluorescent polyphenol, it can be applied as a sensing material for chemical detection in foodstuffs. Curcumin consists of 1,3-diketones which act as an effective scavenger molecule for free radicals, and it can undergo keto-enol isomerization and complexation by chelating with heavy metal ions [9]. Besides, it exhibited substantial chelating potential for Ferrum (II) and Ferrum (III) ion detection, whereas this has sparked much attention [10]. Green banana Saba (*Musa acuminata* x *balbisiana*) is extensively grown in tropical and subtropical regions. It is known as one of the best starch sources because it is readily available in local markets, and approximately 36% of starch content can be found in unripe bananas [11,12]. Besides, banana starch is also well-known for its high resistance to hydrolysis and containing higher amylose content than other starch sources from potato, corn, and wheat [13]. Aloe vera gel is a complex matrix with bioactive properties and a colorless mucilaginous fraction obtained from the parenchymatous cells of *Aloe* spp. fresh leaves [14]. The aloe vera gel consists of approximately 99.5% water and 0.5% polysaccharides, mainly cellulose, hemicellulose, vitamins, minerals, organic acids, phenolic compounds, glucomannan, and mannose derivatives [15]. The aloe vera has been intensely interested in its unique edible coating in food industries to extend the post-harvest storage of fruits and vegetables due to its antifungal properties [16]. Besides, aloe vera gel has been found as one of the best biodegradable films due to its properties such as film-forming, antimicrobial, and biochemical [17]. 

In heavy metal ion detection, conventional techniques have been implemented as the effective known method to determine the presence and concentration of heavy metal ions. The most common conventional methods utilized for the identification of heavy metals include atomic absorption/emission spectrometry (AAS/AES), hybrid generation atomic absorption spectrometry (HG-AAS), inductively coupled plasma optical emission spectrometry (ICP-OES), inductively coupled plasma mass spectrometry (ICP-MS) and high-performance liquid chromatography (HPLC) [9]. These approaches possess high sensitivity and selectivity in identifying the presence of various heavy metal ions. However, a few significant drawbacks include being expensive, time-consuming, requiring complex equipment and skilled employees, and being a multi-step procedure and making this unsuitable for on-site analysis. In other words, designing a simple, cheap, and rapid heavy metal detection technology necessitates a high level of environmental safety. Previously, Werner [18] detected heavy metals of Pb2+, Cd2+, Co2+, Cu2+, and Ni2+ in river and lake water samples using high-performance liquid chromatography combined with a UV detector (LC-UV). Besides, Xing et al. [19] also determined trace metals (Cd2+, Co2+, Cu2+, Mn2+, Ni4+, Pb2+, and Zn2+ in bottled water and tap water samples using inductively coupled plasma mass spectrometry (ICP-MS) and high-performance liquid chromatography (HPLC).

Antecedently, many researchers have extensively exploited nanocomposite films to entrapped colorimetric reagents to detect Fe^2+^. Recently, Choodum et al. [20] designed a biodegradable sensor entrapping 1,10-phenanthroline onto tapioca starch for ferrous detection in soil and water samples with a detection limit of 0.09 ± 0.01 mg L^−1^. The clear film turned red when it was exposed to a ferrous solution. Besides, Kumar et al. [21] immobilized curcumin particles onto a nanoprobe of zeolitic imidazolate framework-8 (Cmim@ZIF-8) to detect Fe^2+^ ions in an aqueous medium with a detection limit of 7.64 μM. Due to the displacement of the curcumin indicator from the surface of ZIF-8, Cmim@ZIF-8 exhibits a decrease in fluorescence intensity in the presence of Fe^2+^. In addition, Sun et al. [22] synthesized a colorimetric and fluorimetric sensor for Fe^2+^ detection using carbon dots and a phenanthroline system (mPD-CDs) with a detection limit as low as 0.59 μM. In the presence of Fe^2+^, the clear solution of mPD-CDs progressively became rufous, and the color intensified until it could be seen by the naked eye. Previously, Liu et al. [23] constructed a rapid and high efficiency of carbon dots derived from acrylamide and chitosan to detect Fe^2+^, and the limit of detection was as low as 160 nM. The color shifted from blue to green when reacted to Fe^2+^. Instead of employing nanotechnology for Fe^2+^ detection, our research has focused on utilizing biopolymer as the carrier for curcumin. 

Recently, there has been an increasing interest in developing optical sensors or indicators with high sensitivity and selectivity by using curcumin as a necessary reagent for colorimetric detection while other natural biopolymer-based materials as carriers [24,25]. The encapsulation could be a new strategy by fabricating curcumin particles within a matrix or shell material to form a capsule thus, overcome the limitations of curcumin and thereby increasing the bioavailability, solubility, and stability of curcumin. It tends to improve the effectiveness of curcumin to chelate the heavy metal ion, hence increasing the sensitivity and selectivity of curcumin-based sensors. Starch films could be used as matrix or shell material to encapsulate curcumin particles because starch is abundant in nature, renewable, and capable of forming a continuous matrix. According to previous research reported by Ortega-Toro et al. [15], starch films have some disadvantages compared to films made of conventional synthetic polymers and other biopolymers. This includes high water solubility, poor mechanical properties, and poor water vapor barrier due to the hydrophilicity properties, leading to the limitations in application with high moisture content. 

Conventionally, this research utilized starch from Green banana Saba (*Musa acuminata* x *balbisiana*) and aloe vera gel as the carrier for curcumin particles to detect the presence of Fe^2+^ ions. Banana starch shows high starch content and is suitable to be employed as a film matrix. Nevertheless, banana starch possesses some limitations that cause the film to have poor film properties. The incorporation of aloe vera in the current research counterbalanced the drawbacks of the starch film resulting in enhanced film properties. Curcumin, an iron chelator, is implemented on the film matrix to detect the presence of Fe^2+^ with high selectivity towards divalent cation of Fe^2+^ due to the selectivity effect on proteins of iron metabolisms that is distinctive from other heavy metals [21]. Therefore, the composite film is a promising tool to be extensively used to monitor Fe^2+^ heavy metals in the environment, especially water and agricultural products, using a film entrapped with curcumin particles.

## 2. Materials and Methods

### 2.1. Materials

The green banana Saba (*Musa acuminata* x *balbisiana*), turmeric powder, and Aloe Vera leaves (*Aloe barbadensis*) were obtained from Kota Kinabalu, Sabah, Malaysia. Citric acid was bought from Sigma Aldrich, St. Louis, MO, USA. Ammonium ferrum (II) sulfate hexahydrate, nickel (II) sulfate hexahydrate, plumbum (II) nitrate were obtained from Sigma Aldrich, St. Louis, MO, USA. All reagents were analytical grade and prepared using Milli-Q water.

### 2.2. Fabrication of Aloe Vera-Banana Starch-Curcumin Composite Film

Green bananas were peeled, sliced into 2-cm slices, and immersed in a citric acid solution (2% *w*/*v*) for 5 min and blended for 2 min. The banana pulp was filtered while being washed through filters, resulting in distilled water free of solutes and suspended solids. The solution was left for 2 h before the supernatant was gently removed [26]. The residual solution was centrifuged for 15 min at 3000 rpm at room temperature, and the supernatant was discarded again. The white starch sediments were dried for 18 h in a universal oven at 55 °C and sieved with aperture sizes of 850 mic and 250 mic [27]. The aloe vera leaves were washed and dried. After that, the outer green skin layer was removed, and the aloe vera gel was collected. The aloe vera gel was washed, dried, and blended for 3 min. Then, the blended pulp was filtered using a sieve to discard solid residues arising from cell walls. The aloe vera extract was pasteurized at 70 °C for 45 min and cooled at room temperature to prolong the shelf life. The aloe vera extract was stored at 4 °C until required. 

The turmeric stock solution was prepared by dissolving the turmeric powder in acetone (1:20). The stock solution was sonicated by an ultrasonic power with a frequency of 40 kHz at 35 °C for 30 min. After sonication, the solution was filtered, and the orange-colored solution was obtained and evaporated at 40 °C [28]. The banana starch was mixed with aloe vera extracts with a ratio of 5:30. About 0.2% (*w*/*w*) of curcumin extracts were added when needed. The suspension was heated with constant magnetic stirring at 60–75 °C for about 45 min. The suspension was homogenized until a yellowish-brown viscous product was formed. Finally, 3 mL of the suspension was cast onto Petri dishe’ bottom and dried in the laminar airflow at room temperature for further use [29].

### 2.3. Morphology Characterization 

SEM (JEOL JSM-5610LV, JKA Technical Cooperation, Tokyo, Japan) was used to assess the morphological structure of the generated composite film. The sample was fixed on aluminum stubs using double-sided tape and coated with a platinum layer to improve conductivity. The coated samples were observed under SEM using an acceleration voltage of 10 kV [30].

### 2.4. Film Thickness and Mass

The thickness of the film was measured with slight adjustments using the method published by Nieto-Suaza et al. [29]. A micrometer with 0.01 mm accuracy is used to evaluate the film thickness (mm). The mean value was computed after five measurements were performed at random spots on each film. An electronic balance was used to determine the film mass (g).

### 2.5. Density 

The density was calculated using a slightly modified version of the approach given by Rovina et al. [31]. The film was divided into (2 cm × 2 cm) pieces, and the original weight was estimated. The film was dried for 24 h at 105 °C in a universal oven while the final weight was measured and recorded (W_f_). The density was evaluated using Equation (1).
(1)$$Density=\frac{Wi-Wf}{A\times x}$$

W_i_ is the initial weight of film before drying (g), W_f_ is the final weight of film after drying (g), A is the area of the film (cm^2^), and x is the film thickness (mm).

### 2.6. Color Property

The color of the films is assessed using a hand-held colorimeter (Konica Minolta, Tokyo, Japan) in absorption mode by using CIE Lab-scale (L*, a*, and b*). The chromaticity parameters a* (green [-] to red [+] and b* (blue [-] to yellow [+]) are used to represent the brightness L* (L* = 0 for black and L* = 100 for white). The films were measured in triplicate, and the total color difference was determined using Equation (2).
(2)$$\Delta E=\sqrt{{\left(L_{2}^{*}-L_{1}^{*}\right)}^{2}+{\left(a_{2}^{*}-a_{1}^{*}\right)}^{2}+{\left(b_{2}^{*}-b_{1}^{*}\right)}^{2}}$$

L_1_*, a_1_*, and b_1_* are color values of the standard white plate, while L_2_*, a_2_*, and b_2_* are the color values of sample films.

### 2.7. Opacity

The method provided by Bojorges et al. [32] was used to assess the film’s opacity. The UV-Visible Spectrophotometer (PerkinElmer, London, UK) was used to analyze the absorbance of film at 600 nm. A film was cut (4 cm × 1 cm) and placed in a cuvette. The measurements were taken in triplicate while the opacity of the film was estimated using Equation (3).
(3)$$\;Opacity=\frac{A_{600}}{x}$$

A_600_ is the absorbance at 600 nm, and x is the film thickness (mm).

### 2.8. Biodegradation Tests

The generated composite film is biodegraded in soil, with slight modifications to the technique reported by Zhou et al. [25]. The soil was layered in a plastic tray to a thickness of about 4 cm. The films were divided into square pieces (2 cm × 2 cm) and buried 2 cm below the soil’s surface. The plastic tray’s initial weight (M_0_) buried with the film was measured and kept at ±25 °C. The plastic tray containing the deteriorating samples and fragments was weighed every day for 5 days. The biodegradation percentage of the finished film was calculated using Equation (4).
(4)$$\;Biodegradation\;\%=\left[\frac{\left(M_{0}-M_{1}\right)}{M_{0}}\right]\times 100\%$$

M_0_ is the initial weight of the plastic tray buried with the original films before the biodegradation test (g), and M_1_ is the weight of the plastic tray buried with the residual films after the biodegradation test (g).

### 2.9. Moisture Content

The moisture content of the films was specified using the method represented by Lee et al. [33]. Films were cut into square specimens of 2 cm × 2 cm, and initial weight was measured. Subsequently, the films were dried in a universal oven at 105 °C for 24 h. The moisture content was calculated using Equation (5).
(5)$$\;Moisture\;content\;\left(\%\right)=\frac{\left(W_{w}-W_{d}\right)}{W_{w}}\times 100$$

W_w_ is the initial weight of the film (g), and W_d_ is the constant weight of the film after drying at 105 °C for 24 h (g).

### 2.10. Water Solubility 

The water solubility of each film was determined as described by Wang et al. [34] with modification. Each film with a size of 2 cm × 2 cm was dried at 105 °C for 24 h to obtain a constant weight and was recorded as the initial weight. Each sample was then immersed in 50 mL of ultrapure water and kept at room temperature for 24 h. After hydration, the wet film was gently wiped using the filter paper and dried in an oven at 105 °C for approximately 24 h to a constant weight with good accuracy and reliability of data. Therefore, the water solubility (%) of the film was determined using Equation (6).
(6)$$\;Water\;solubility\;\left(\%\right)=\frac{\left(Wi-Wf\right)}{Wi}\times 100$$

W_i_ is the initial weight of the film sample (g), and W_f_ is the final weight of the dried sample after water immersion (g).

### 2.11. Water Absorption

Water absorption of the film prepared was determined using the procedure described by Rovina et al. [31] with slight modification. To obtain the constant weight, three replicates of films were prepared and dried overnight at 105 °C to obtain the constant weight with good accuracy and reliability of data. The initial dry weight was recorded as W_t=0_. The dried films were placed into containers with a relative humidity of approximately 57%. The controlled temperature and pressure were fixed at 25 °C and 1 atm throughout this analysis. The final weight of films after 24 h in a stationary state was measured and recorded as W_t=24_. The water absorption of the film was calculated using Equation (7).
(7)$$\mathrm{Water}\;\mathrm{absorption}\;(\%)=\frac{W_{t=24\;h}-W_{t=0\;h}}{W_{t=0h}}\times 100\%$$

W_t=oh_ is the film’s initial dry weight before storage (g), and W_t=24 h_ is the film’s weight after 24 h (g).

### 2.12. Swelling Degree

The swelling degree of film was determined using the method described by Wang et al. [34] with some modifications. Each film with a size of 2 cm × 2 cm was immersed in 50 mL of ultrapure water and kept at room temperature for 24 h. After hydration, the swollen film was gently wiped using the filter paper and weighed 24 h (W_s_). Then, the film was dried in an oven at 105 °C for approximately 24 h to a constant weight (W_d_). The swelling degree of film was defined using Equation (8).
(8)$$\;Swelling\;degree\;\left(\%\right)=\frac{W_{s}-W_{d}}{W_{d}}\times 100\%$$

W_s_ is the swollen film weight (g), and W_d_ is the dried film weight (g).

### 2.13. Water Vapor Permeability

Water vapor permeability values were determined by using the American Society for Testing and Materials (ASTM) E96 method, known as the “cup method” [35]. ASTM E96 was the standard test for gravimetric determination of water vapor permeability rate. A wide-mouth plastic cup with a diameter of approximately 9 cm was filled with 50 g of silica gel. The film was sealed on the mouth of the cup by using cellophane tape and aluminum foil. The cup was then placed at 20 °C inside a desiccator with distilled water. Weight was measured each 24 h for 5 days. The del plot of weight vs. time slope was calculated by linear regression (r^2^ > 0.99). Water vapor transmission rate (WVTR) was calculated as the ratio slope (g/s)/test area (m^2^). Water vapor permeability (WVP) was calculated according to the combined Fick and Henry laws for gas diffusion through films, using Equation (9).
(9)$$WVP=WVTR\times \frac{x}{\Delta P}$$

x (m) is the film thickness, and ΔP (Pa) is the differential of water vapor pressure through the film. A driving force of 2339 Pa was used as the differential vapor pressure of water.

### 2.14. Statistical Analysis

The data obtained were analyzed by IBM SPSS Statistics 27. IBM SPSS Statistics is a statistical software platform used to solve research problems using hypothesis testing, predictive analytics, ad hoc analysis, and geospatial analysis. The one-way analysis of variance (ANOVA) and Tukey’s test were performed to determine the differences between the properties of films in the 95% confidence range. Tukey’s test was run to confirm the differences between the properties of the banana starch film, aloe vera-banana starch film, and aloe vera-banana starch-curcumin film.

## 3. Results and Discussion

### 3.1. Morphological Characterization

The surface and cross-section images of the developed films by SEM are shown in Figure 1. Based on Figure 1a, the pure banana starch films showed smooth and homogeneous surfaces, which may be related to their high content of starch, especially amylose, and a small content of other macromolecules. The presence of starch molecules formed stable complexes in the polymer matrix, resulting in a homogeneous polymeric structure of the banana starch film [36]. The banana starch films added with aloe vera gel presented a rougher surface shown in Figure 1b. The presence of aloe vera gel caused the aggregation of components owing to a high solid content of aloe vera gel, leading to the formation of dispersed fine clumps with rough surfaces [36,37]. However, Nieto-Suaza et al. [29] reported that the inclusion of aloe vera gel in the filmogenic solution of banana starch does not affect the crystallinity of banana starch film due to the excellent cross-linking interaction between phenolic compounds of aloe vera gel and starch molecules. 

The films added with curcumin displayed a rough surface with slightly aggregated particles represented in Figure 1c,d. The tiny particles with irregular crystal shapes were observed on the banana starch-curcumin film and aloe vera-banana starch-curcumin film. Curcumin particles promoted discontinuity of the starch matrix, causing irregularities on the surface of the film. Surface morphology in Figure 1e–h revealed that these crystalline particles of curcumin were randomly distributed throughout the surface, while some of them were aggregated within the film, resulting in the increased roughness of the film surface. The curcumin were dispersed homogeneously within the film matrix formed by banana starch and aloe vera gel, which was linked to the addition of the bioactive agent in the gelatinization process, allowing a better dispersion in the polymer matrix [38]. The pure carbohydrate-based films showed a smooth and compact surface without cracks, but the incorporation of curcumin led to a slightly rough surface of films. The surface morphology of the composite film could be influenced by the concentration of curcumin incorporated into the film. The films containing both the curcumin and aloe vera gel formulations presented higher heterogeneity than the others, which can be visualized in Figure 1i–l. It could be related to the action of aloe vera gel that aggregates and attaches the crystal-shaped curcumin with banana starch and other polysaccharides together through cross-linking in the film matrix. The organic acids from aloe vera gel, such as citric acid, caused the effect of cross-linking and improved the plasticizing properties of bio-based films [39]. It could improve the hydrophobic nature of the film surface due to the higher intermolecular hydrogen bonding interactions between aloe vera and banana starch, leading to a composite film with higher compactness morphology and more robust water barrier properties [40]. Throughout this, organic acids from aloe vera gel may affect the physicochemical properties of banana starch by triggering crosslinking reactions between starch chains. Consequently, the implementation of curcumin onto the composite films also improved the heterogeneity of the films via cross-linking interactions between the molecules. 

### 3.2. Dry Weight, Density, and Thickness

Table 1 summarizes the values for thickness, mass, density, surface color, opacity, and biodegradability of films. The films produced had a dry surface area of 63.62 cm^2^ and a thickness ranging from 16.80 μm and 24.40 μm. Film thickness usually plays a fundamental role in determining the film’s barrier and mechanical properties, as well as regulating the viability of employing the film as packaging materials or secondary protective materials to carry the sensitive reagent. Aloe vera-banana starch-curcumin films had significantly enhanced film thickness of approximately 9.20 μm when compared to control films. This demonstrated that the film thickness could be enhanced by the addition of aloe vera gel, which is attributed to the cross-linking effect of the polyphenolic chemicals found in aloe vera AV gel with starch molecules. Besides, curcumin also plays a role in enhancing the thickness of the film via polymeric interaction between curcumin and starch molecules [41]. Thus, the addition of aloe vera gel and curcumin particles can significantly increase the thickness of the banana starch edible films [40,42]. 

### 3.3. Color Property and Opacity

Color is one of the most essential aspects of a composite film’s visual appearance, and it has a direct impact on the film’s color-changing ability as a colorimetric sensor. Table 1 shows that the control banana starch film had the lightest color. Polysaccharide-based films are typically colorless and are preferred because the color of the detection reagent is less likely to be affected and the color changing of the film is easy to detect. The addition of aloe vera gel significantly increased the a* value, indicating the presence of characteristic tones of red color in aloe vera-banana starch-curcumin film compared to pure banana starch film and decreased the lightness of the film [42,43]. The changes in the color of the film caused by the addition of curcumin extract could be related to the color of the plant extract itself or the binding of extract to the polymer, and it will vary depending on the type and concentration of the extract used. The overall color varied between each film made with a different formulation. The banana starch and aloe vera-banana starch composite films with curcumin revealed a bright yellow color due to the presence of curcumin [44]. The inclusion of curcumin contributes to an intense orange color to the composite film. The film opacity values increased from 35.19 ± 2.37 to 38.01 ± 4.51 as curcumin was added to the banana starch film, indicating that the inclusion of curcumin reduced the film transparency due to the increasing solid content in the filmogenic solution [29]. However, when curcumin was added to the composite film in the presence of aloe vera gel, the opacity value of the film decreased. It could be explained by the ability of the aloe vera gel to disperse curcumin in the filmogenic solution. The aloe vera content might influence the degree of crystallinity of amylose in starch films, affecting the film’s optical behavior [15].

### 3.4. Biodegradability Test

The biodegradation test, also known as the soil burial test, is used to determine the biodegradability of the developed composite film to the environment by measuring the residue weight lost in the soil within a week. Table 1 shows the biodegradability of films made of different material compositions. The soil burial test was conducted for four days in order to establish the biodegradability capabilities of the produced films. The weight losses were recorded each day for all bio-films during the four days of the soil exposure test. The weight loss of each film was primarily due to the biodegradation by microorganisms and their enzymes found in the soil. The bacteria from genera such as Sphingomonas, Bacillus, Pseudomonas, Achromobacter, and Mycobacterium were involved in the process of biodegradation [45]. The biodegradability of bio-films ranged from 0.79% to 1.28%, but the differences between the films were not statistically significant (*p* > 0.05). It could be explained that bio-films were highly degraded and lost their original form [46]. The presence of hydrolyzable and oxidizable linkages such as enol-ketone, benzene ring, and hydroxyl group in the polymer chain increases films’ biodegradability rate. The morphological structure of the polymers allows easy access of microorganisms and enzymes to the polymer chain, thus increasing the degradation rate of the developed biofilm [46].

### 3.5. Water-Resistance/Water Barrier Properties 

Water resistance, also known as water barrier property, refers to the ability of biofilms to resist water vapor and liquid water. It shows the film’s capacity to resist moisture absorption from the higher concentration side to the lower concentration side. It is one of the most important features for bio-film applications as detection films since it affects the storage stability and efficiency of the films. To determine the water barrier property of the film, the moisture content, water-solubility, water absorption, swelling degree, and water vapor permeability of bio-films were measured. The moisture content of composite films is generally contributed by the number of water molecules available in the network microstructure of films. Referring to Table 2, the moisture content of the four films does not significantly differ. The films containing aloe vera gel contain higher moisture content compared to the control film. It was contributed by the aloe vera gel consisting of 90% of water content. However, incorporating curcumin negligibly reduces the moisture content of bio-films because the curcumin particles improve the hydrophobicity of the composite films by intermolecular interaction through hydrogen bonding between the components [47]. The bioactive compound curcumin can reduce film’s moisture content due to its low affinity with water, promoting film humidity reduction [38]. Our previous study also mentioned that increased curcumin concentration inhibits moisture transfer and enhances the hydrophobic characteristics of the films [41].

Water solubility and water absorption are the integral factors determining the films’ water resistance. Table 2 depicts the water solubility of films after immersing in the ultrapure water for a day. The significantly improved water solubility at room temperature was observed in the films containing aloe vera gel, with a solubility of 11.70% and 9.45% for the aloe vera-banana starch film and aloe vera-banana starch-curcumin film, respectively. The result was similar to the findings of Kanatt & Makwana [42], where adding aloe vera gel can increase the solubility of the film. It can be related to the hydrophilic nature of aloe vera gel, which promotes the diffusion of water molecules into the film matrix and thus increases the water solubility by weakening the interactions between the molecular chains of the polymers and increasing the free volume between the chains [48]. Besides, a reduction of water solubility was observed in the films with curcumin, but the difference was not statistically significant (*p* > 0.05). These findings were also agreed with the research done by Xu et al. [49]. The incorporation of curcumin reduced the water solubility of the banana starch film due to the hydrophobic property of curcumin that limited water interaction with the polymeric matrix. Moreover, the water absorption of bio-films ranged from 27.38% to 43.65%. The swelling degree of swelling capability of the film is defined as the ability of a film to trap water molecules. The swelling degree of four bio-films ranged between 118.75% and 157.84%, but the difference was statistically insignificant. The addition of the aloe vera gel to the film increased the degree of swelling to 157.56% and 157.84%, following the findings of Kanatt & Makwana [42].

### 3.6. Sensing Performance of the Composite Film

Film sensing ability is measured via the determination of the sensitivity of the developed colorimetric sensor towards different concentrations of Fe^2+^ standard solution and its selectivity towards different types of standard metal solutions. The prepared solution for metals includes Fe^2+^, Fe^3+^, Ni^2+^, Cd^2+^, Hg^2+^, and Pb^2+^ solutions. A series of standard Fe^2+^ solutions with concentrations from 0 to 100 ppm are prepared. The absorbance of the standard Fe^2+^ solutions is measured between 200 and 600 nm to get the spectra with a bandwidth of 2 nm. The highest absorption is observed at 465 nm, and hence it is selected as the maximum wavelength for quantification of Fe^2+^. Figure 2a shows the calibration graph of absorbance against the standard Fe^2+^ solution concentration with the equation of y = 0.0064x + 2.259 and the slope of the curve, which is 0.0064, where x represents the concentration of Fe^2+^ ion standard solution, and y represents the absorbance of Fe^2+^ ion standard solution. When the color of the developed colorimetric sensor changed from yellow to greenish-brown, the qualitative analysis was determined. Simultaneously, the UV-Visible Spectrometer was used to do quantitative analysis and determine the absorbance of the Fe solution. All measurements were made in quintuplicate, and the average concentration with its standard deviation (Sa) were calculated. The limit of detection (LOD) and limit of quantitation (LOQ) were determined using the equation described by Shrivastava & Gupta [50], which are LOD = 3Sa/b and LOQ = 10Sa/b, where Sa is the standard deviation of the response and b is the slope of the calibration curve. The calibration curve relationship between absorbance and concentration showed a highly linear response with R2 = 0.9845. The limit of detection (LOD) and limit of quantification (LOQ) values are 27.8438 ppm and 92.8125 ppm, respectively. The developed film performances, including detection range and detection limit, are almost comparable to previously reported composite materials (Table 3). 

Based on the photographs as shown in Figure 2b, the composite film suspension mixed in the ratio of 1:10 with Fe^2+^ ion standard solution changed its color from yellow to greenish-brown. The brown color intensity of the film suspension increased as the metal ion standard solution concentration increased and reaction time increased. The colorimetric detection of Fe^2+^ with the aloe vera-banana starch-curcumin composite film is based on the reaction of the curcumin added to the sensor with the Fe^2+^ ion solution at different standard concentrations ranging from 0 to 100 ppm. After the composite film suspension was immersed in Fe^2+^ ion solution, the curcumin in the film chelated to the ferrum to form a greenish-brown complex within the composite film, causing the color change from yellow to greenish-brown, which allows colorimetric detection of Fe^2+^ present in the solution. The mechanism of formation is shown in Figure 3. The interaction between ferrum and curcumin has been recognized in studies reported by Beneduci et al. [51]. As mentioned, curcumin dissolved in ethanol strongly absorbed the UV–vis region at 426 nm, whereas the mixture of film suspension containing curcumin and Fe^2+^ ions solution showed the maximum absorption of 410 nm. There was a left shift in the absorption wavelength of curcumin from 426 nm to 410 nm observed using UV–Vis spectroscopy, which confirmed the formation of curcumin-Fe^2+^ complexes that led to the color changes of the film. The phenomenon of the color change of the curcumin-based film to identify the presence of metal ions and other contaminants was also agreed with other studies reported by authors, including Rasouli & Ghavami [24] and Zhou et al. [25]. As shown in Figure 2b, when the concentration of standard Fe^2+^ ion solution increased from 0 to 100 ppm, the greenish-brown color of the aloe vera-banana starch-curcumin composite film became darker due to the increase in the formation of ferrum-curcumin complex within the film [52]. 

### 3.7. Selectivity of the Composite Film

The selectivity of the aloe vera-banana starch-curcumin film was tested with various metal ions such as Fe^2+^, Fe^3+^, Pb^2+^, Ni^2+^, Cd^2+^, and Cu^2+^ ions at 100 ppm. Based on Figure 4a, color changes seen by the naked eyes only occurred for the composite film suspension immersed in the Fe^2+^ solution, which turns from yellow to greenish-brown. However, the other metal ions did not show color change and remained the yellow color. The color change in Fe^2+^ ion was due to the formation of the curcumin complex, which was confirmed using UV–Vis spectroscopy with a left shift in the absorption wavelength of curcumin from 426 nm to 410 nm, as mentioned before. Based on Figure 4b, it can be seen that Fe^2+^ has the highest absorbance reading and shows a reaction towards curcumin. This happens due to curcumin acting as an iron chelator with high selectivity towards divalent cation of Fe^2+^, which is caused by the selectivity effect on proteins of iron metabolisms that is distinctive from other heavy metals [34]. Hence, Fe^2+^ ions can bind to curcumin to form a ferrum-curcumin complex.

## 4. Conclusions

In this study, the developed bio-film of green banana starch and aloe vera gel has successfully integrated with curcumin to generate a colorimetric sensor for Fe^2+^ ions detection, where the yellow filmogenic solution turned greenish-brown in the presence of Fe^2+^ ions. As the concentration of Fe^2+^ solution and reaction time increases, the color intensity of aloe vera-banana starch-curcumin composite film increases. The following synergism results from curcumin-Fe^2+^ complex formation contributed to the greenish-brown color in the composite film and can be seen through the left shift of absorption wavelength in UV-Vis spectroscopy outcome. The designed composite films showed high selectivity towards Fe^2+^ ions within a brief response time. The effective extraction of starch from green banana Saba and the incorporation of curcumin extracts and aloe vera gel for the bio-film were the novelty of the current study. Banana starch possesses some drawbacks, which have been counterbalanced with the inclusion of aloe vera gel to enhance the properties of the film. Comparatively to other natural biopolymers, the film behavior was successfully revealed to have less transparency, thus, providing a distinctive color change when interacting with Fe^2+^ ions. Besides, the proposed composite films showed good water vapor permeability and allowed only Fe^2+^ ions to react with curcumin particles. At the same time, all the developed films also exerted to be ingeniously biodegradable due to the native properties of the plants. The composite film showed a favorable bio-film to be widely used in the food and agriculture industry to detect the presence of Fe^2+^ in real samples with great satisfactory sensing performance towards Fe^2+^ ions.

## Figures and Tables

**Figure 1 polymers-14-02353-f001:**
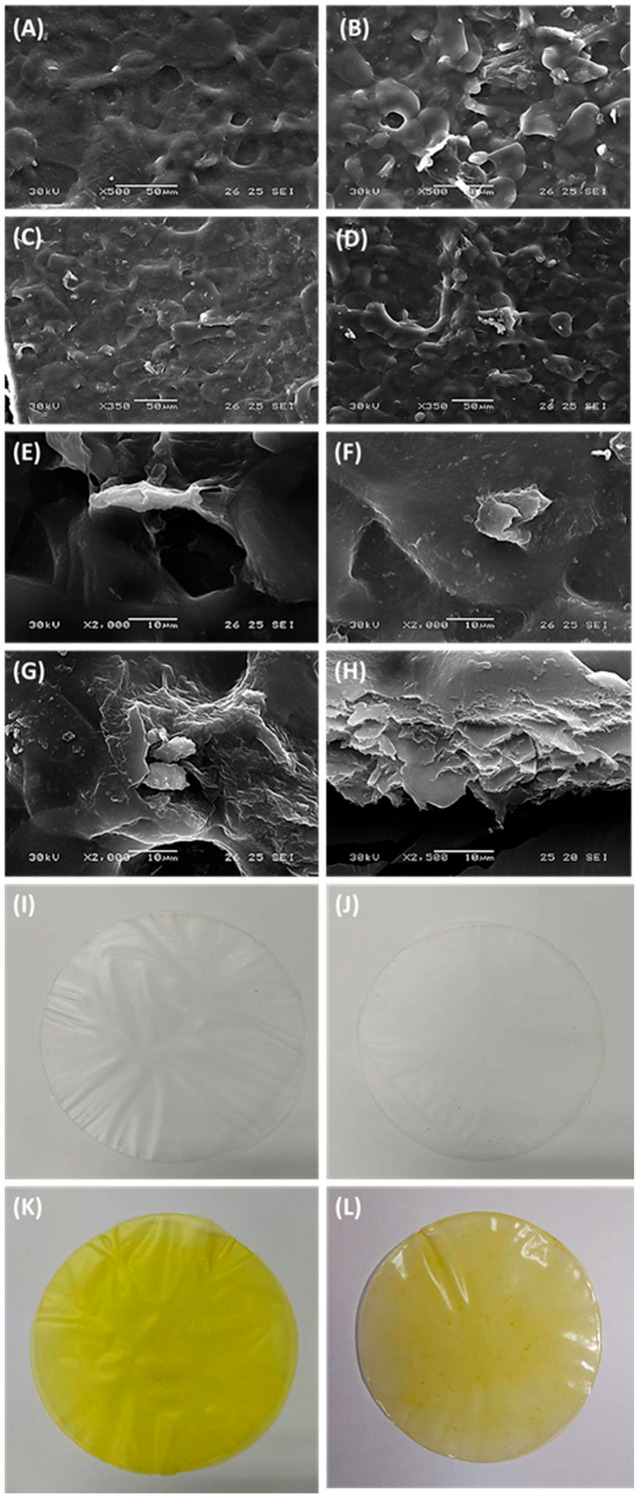
The SEM images of (**A**) banana starch film, (**B**) aloe vera-banana starch film, (**C**) banana starch-curcumin film, and (**D**) aloe vera-banana starch-curcumin film at 50 μm magnification. The SEM images of the interaction between (**E**) banana starch and aloe vera, (**F**) banana starch and curcumin, and (**G**) aloe vera with banana starch and curcumin at 10 μm magnification. (**H**) The cross-section image of aloe vera-banana starch-curcumin film at 10 μm magnification. Surface appearance of (**I**) banana starch film, (**J**) aloe vera-banana starch film, (**K**) banana starch-curcumin film and (**L**) aloe vera-banana starch-curcumin film with a diameter of 90 mm.

**Figure 2 polymers-14-02353-f002:**
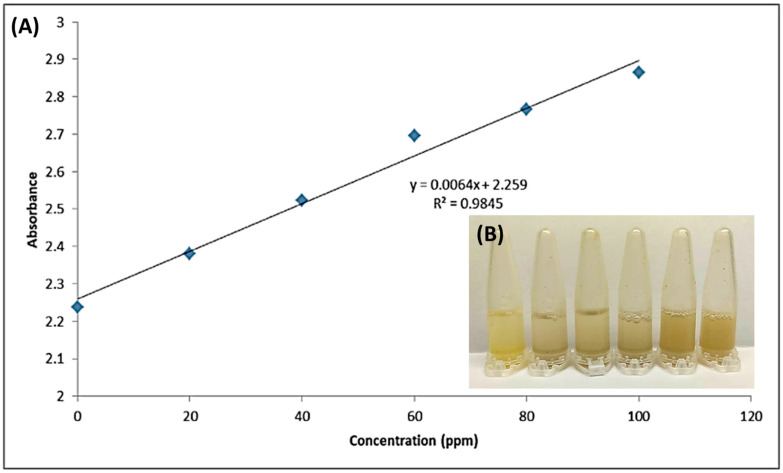
The (**A**) calibration curve of absorbance at a wavelength of 410 nm and (**B**) photograph shows different colors of composite film suspension added with Fe^2+^ ions at different concentrations against metal ion standard solution concentration in the range of 0 to 100 ppm at a wavelength of 410 nm (Conditions: ± 25 °C, 45 ± 5% RH; Response time: 10 s).

**Figure 3 polymers-14-02353-f003:**
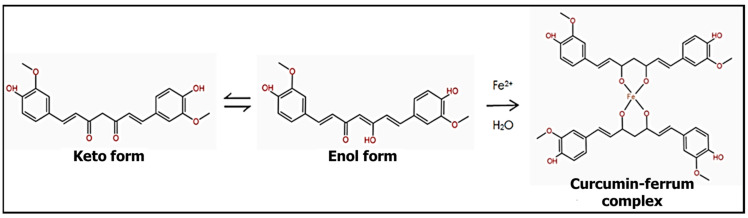
Formation of curcumin/ferrum complex.

**Figure 4 polymers-14-02353-f004:**
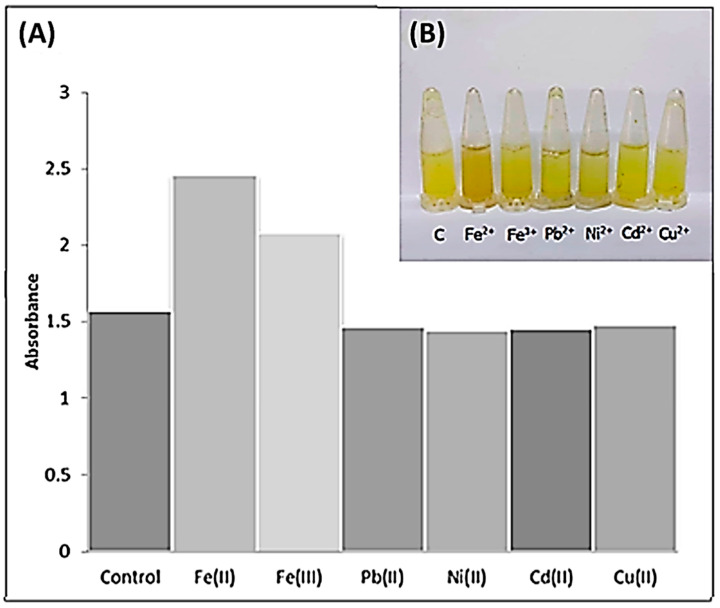
(**A**) The image of the composite film suspension that was mixed in the ratio of 1:10 with different types of metal ion solutions with 100 ppm concentrations (Conditions: ± 25 °C, 45 ± 5% RH; Response time: 10 s) and (**B**) the absorbance of composite film suspension added with metal ions.

**Table 1 polymers-14-02353-t001:** Thickness, mass, density, surface color, opacity, and biodegradability of films.

Parameter	Banana Starch Film	Aloe Vera-Banana Starch Film	Banana Starch-Curcumin Film	Aloe Vera-Banana Starch-Curcumin Film
Thickness (μm)	16.80 ± 0.70 ^a^	26.10 ± 3.30 ^b^	17.40 ± 1.80 ^a^	26.40 ± 1.90 ^b^
Weight (g)	0.18 ± 0.01 ^a^	0.20 ± 0.02 ^a^	0.20 ± 0.01 ^a^	0.23 ± 0.03 ^a^
Density (g/cm^3^)	0.06 ± 2.71 ^a^	0.07 ± 15.04 ^a^	0.08 ± 14.60 ^a^	0.08 ± 1.99 ^a^
ΔE	19.34 ± 0.25 ^a^	22.16 ± 0.14 ^b^	52.07 ± 0.85 ^c^	42.38 ± 0.67 ^d^
Opacity (%)	35.19 ± 2.37 ^a^	38.22 ± 4.62 ^a^	38.01 ± 4.51 ^a^	32.77 ± 2.10 ^a^
Biodegradability (%)	1.28 ± 0.05 ^a^	1.18 ± 0.11 ^a^	0.79 ± 0.76 ^a^	1.28 ± 0.03 ^a^

Different lowercase letters in the same column indicate a statistically significant difference (*p* < 0.05).

**Table 2 polymers-14-02353-t002:** Water-resistance properties of films.

Parameter	Banana Starch Film	Aloe Vera-Banana Sarch Film	Banana Starch-Curcumin Film	Aloe Vera-Banana Starch-Curcumin Film
Moisture content (%)	19.16 ± 2.71 ^a^	27.43 ± 15.04 ^a^	14.56 ± 0.39 ^a^	27.12 ± 1.42 ^a^
Water solubility (%)	3.97 ± 0.18 ^a^	11.70 ± 4.22 ^b^	1.75 ± 0.96 ^a^	9.45 ± 1.24 ^b^
Water adsorption (%)	27.38 ± 3.87 ^a^	38.11 ± 21.88 ^a^	27.37 ± 19.50 ^a^	43.65 ± 1.75 ^a^
Swelling degree (%)	145.91 ± 31.32 ^a^	157.84 ± 103.77 ^a^	118.75 ± 6.77 ^a^	151.56 ± 52.14 ^a^
Water vapor permeability (g/m s Pa)	8.05 × 10^−7^ ± 7.4910^−9 a^	5.3010^−7^ ± 2.4910^−8 a^	5.5910^−7^ ± 2.6310^−8 a^	3.3810^−7^ ± 1.8010^−8 a^

Different lowercase letters in the same column indicate a statistically significant difference (*p* < 0.05).

**Table 3 polymers-14-02353-t003:** Comparison of different composite materials to determine Fe^2+^ ions.

Composite Materials	Linear Range	LOD	Samples	References
Carbon dots with acrylamide/ chitosan	0–50 μM	160 nM	Drinking water	Liu et al. [23]
Carbon dots/ phenanthroline	0–100 μM	2.98 μM	Lake water, tap water, and human blood	Sun et al. [22]
Curcumin immobilized zeolitic imidazolate framework-8	0–250 μM	7.64 μM	Aqueous media	Kumar et al. [21]
Phenanthroline/ tapioca starch thin film	0–10 ppm	0.09 ± 0.01 ppm	Soil and paddy field water	Choodum et al. [20]
Aloe vera-banana starch-curcumin bio-films	0–100 ppm	27.8438 ppm	Water	This work

## Data Availability

Not applicable.

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
