# Peer review of "Development of Aloe Vera-Green Banana Saba-Curcumin Composite Film for Colorimetric Detection of Ferrum (II)"

_polymers, 2022, doi:10.3390/polym14122353_

Round 1

Reviewer 1 Report

This paper by Rovina et al presents preparation of composite films from the substances from multiple plants and their use in detection of Fe2+. The study employed local plants as the materials and is unique. This can be potentially publishable in this journal. The following issues are to be addressed in revision of the manuscript.

1) Abstract and throughout the manuscript: the film is mentioned as a bio-degradable film. I recommend the authors to employ the words, bio-film, not bio-degradable film. Biodegradable properties of the film was not investigated in this study.

2) Caption of Figure 5 should contain wavelength of the measurement. Is it also 410 nm?

3) Graphs of Figures 2 and 5 are not clear. The original drawing file should be used in the presentation.

Author Response

Dear reviewer,

The authors would like to inform you that the revised manuscript referred to above has corrected the suggestions and constructive comments of the reviewers and editors. The comments and suggestions highlighted by the reviewers were considered in the revised manuscript. The modifications, additions, and corrections appear in the highlight within the article. 

Thank you very much for your kind consideration. 

Reviewer 2 Report

Major Comments:

This study was performed to develop a biodegradable film composed of Aloe vera (Aloe barbadensis), green banana Saba and curcumin for the detection of Fe2+ ions. The film is synthesized by employing green banana Saba incorporated with curcumin and Aloe vera gel to enhance the sensing performance. The article has some shortcomings in regard to some data analyses and text, and this unique dataset has not been exploited to its full extent.  The results shown in this study are lack of innovation and novelty. The literature review is not strong enough to provide research gaps of study. Sentences and abbreviations should be clearer and better for the understanding of readers. Some figures are blurry, they are very indigent in terms of clarity. The English language used in the manuscript needs improvements, as there are some punctuation and grammatical mistakes throughout the manuscript. Sentences need more clarity and better construction. Some figures need more transparency; special focus is required in labelling the axis and titles. Overall, the paper needs some structural and literature revisions to meet the requirements of the journal. The manuscript does not give an idea about the research gaps and thus the importance of this work and its possible future efficiency. Due to the inevitable flaws, the manuscript strongly needs major revisions.

Introduction:   

The introduction is stubby and deficiency of plenty background information which is incapable to give the reader material of this study. The introduction needs to be more emphasized on the research work with a detailed explanation of the whole process considering past, present and future scope. There is a lot of spelling and grammatical mistakes throughout the manuscript. Why this study is essential? How the present study gives more accurate results than previous studies? It is advised to add a recent works survey about numerous types of novel polymer materials, the disadvantages of composite films. It is also recommended to include some strong updated literature survey about what specific authors intended to do in this study explain clearly with some background and future applications. The last paragraph should be containing the novelty of the work in addition to the work done. Research gaps are not highlighted more clearly, and future applications of this study should be added.

Specific Comments:

  1. Abstract: please focus the abstract on your study and results. Particularly the role of biodegradable films? Authors are required to add more data on the development of biodegradable composite films and their thickness. Moreover, authors should pay more attention to the manuscript on the innovative and scientific results.   
  2. Page 3, line 30-40: “Green bananas were peeled, sliced into 2-cm slices, and immersed in a citric acid solution (2% w/v) for 5 min and blended for 2 min. The banana pulp was filtered while being washed through filters, resulting in distilled water (dH2O) free of solutes and suspended solids.” How would you justify this statement?? Please give the reference of this methodology.
  3. Page 3, line 35: “The AV leaves were washed and dried”. Please change AV to Aloe Vera? No abbreviation should be written. Though if you have to right then add a footer or complete form along with the paragraph.
  4. Page 3, section 2 Materials and Methods: Authors have not mentioned what standard methods have been followed, please add a subsection as “Textural characterizations” and compare FTIR/SEM/XRD techniques/methodology with the following recent studies: Journal of Polymers and the Environment, 2021; 29(1):156-174. Chemical Engineering Journal, 2014;235:100-108. Industrial & Engineering Chemistry Research, 2020;59(51):22092-106. Journal of Molecular Liquids, 2019;293:111442.
  5. Why have authors chosen the Aloe Vera for the biodegradable study instead of any other plant sample??
  6. Page 4, line 15: “The film was dried for 24 h at 105 °C in a universal oven while the final weight was measured and recorded.” Is there any particular reason to take 24 hours for this film to dry? On what basis these were nominated? What will happen if the timing increases or decrease?
  7. Most of the mathematical/reaction equations are not cross-referencing in the text. Please cross-reference each equation properly in the text throughout the manuscript.
  8. Page 4, line 11: “SEM was used to assess the morphological structure of the generated composite film. The sample was fixed on aluminium stubs using double-sided tape and coated with a platinum layer to improve conductivity. The coated samples were observed under SEM using an acceleration voltage of 30 kV.” Please provide a reference for the method of determination SEM? Why should authors have chosen this protocol only??
  9. Page 6, line 10: “Water vapour permeability values were determined by using the ASTM E96 method, known as the cup method; explain ASTM E96 method stands for? Is there any other specific method to determine water permeability?
  10. Page 6, line 36: “The excellent dispersion of Bs to form a homogenous polymeric matrix with a smaller diameter is a good condition for microorganisms consuming the biofilm during biodegradable tests.” Specify the diameter of the polymeric matrix and elaborate microorganisms involved in the biodegradable test?
  11. Page 6, line 25: “The analytical determinations were performed in triplicate using IBM SPSS Statistics 27 (2020) and reporting standard deviations. The one-way analysis of variance (ANOVA) was performed, and Tukey's test was used to determine the differences between the properties of films in the 95% confidence range.” It appears that the SPSS was used, but not described. It may be best to let it be subsumed. What are IBM SPSS statistics? Explain why authors have used the Tukeys test approach?
  12. Page 7, line 17: The organic acids from AV gel caused the effect of cross-linking. It could improve the hydrophobic nature of the film surface due to the higher hydrogen bonding interactions, leading to a composite film with higher compactness morphology and more robust water barrier properties.” Which organic acid effect cross-linking?
  13. The discussion presented is very weak no strong comparison has been made with the literature to support the authenticity of the obtained results. Therefore, the authors are suggested to discuss their results with the following recent research about types of novel material, synthesis methods, membranes and novel polymer materials to make the background and discussion stronger: Polymer Testing, 2020;91:106867. Polymers, 2021;13(14):2307. Journal of Polymers and the Environment, 2021; 29(1):156-174. Journal of Polymers and the Environment, 2021;29: 2598–2608. Materials Science and Engineering: C, 2021;126:112127. Materials Science and Engineering: C, 2021;128:112260. TrAC Trends in Analytical Chemistry, 2020; 132:116066.
  14. Page 9, line 1: Figure 1 should be high resolution coloured to understand the crystalline and rough surface of the film. Moreover, the caption should also be revised.
  15. Page 11, Table 2: “Water resistance properties of films.” Correct the spelling of water vapour in the table; Values should be properly spaced.
  16. Page 11, line 26: “The colourimetric detection of ferrum with the AV-Bs-Cr composite film is based on the reaction of the Cr added in the sensor with the standard Fe (II) ion solution. After the composite film suspension was immersed in standard Fe (II) ion solution, the Cr in the film chelated to the ferrum to form a greenish-brown complex within the composite film, causing the color change from yellow to greenish brown, which allows colourimetric detection of ferrum present in the solution.” What is the standard Fe ion solution? and why authors have chosen Fe or Cr in the film instead of Cu and Cr? Elaborate.
  17. The authors are advised to write the conclusions in a comprehensive way and should contain key values, suitability of the applied method, the major findings and contributions.
  18. References: Very limited scope references. The authors are advised to revise this section, including the latest reference. Please see some suggestions in the specific comments and in the ‘introduction’ section.  

Author Response

(The authors gave the same response as above.)

Reviewer 3 Report

  1. The abstract section should be rewritten and look very general and not informative. The abstract section must be revised to highlight the purpose of this work.
  2. In your paper, please highlight the following:
    why this work has been done,
    what is new about it,
    Highlight the novelty (has such work been done before, if so, why you are doing this work).
  3. All the figures must be clearly arranged. Figures quality/resolution must be improved. Must satisfy journal standard. Figure captions must be elaborated with more experimental conditions.
  4. Figure 3- formation of curcumin-ferrum complex. Its recommended to use same notation for curcumin/ferrum.
  5. Figure 4- captions are not matching with the figures.
  6. Curcumin defined as Cr? In the introduction section.
  7. Authors must tabulate the performance of composite materials with literature.
  8. After discussing each section, must be provided with a short conclusion.
  9. Figure X notations must be uniformly cited in the manuscript.
  10. There are many errors in the paper, so the Authors are encouraged to review the form and the English of the manuscript.
  11. Many space errors/punctuation errors must be solved. Abbreviations must be clearly followed throughout the manuscript. Parameters are defined several times in the manuscript.
  12. Conclusion section- must focus on future directions of these composite materials? General statements need to be removed. Authors are suggested to be more specific in their conclusions.

Author Response

(The authors gave the same response as above.)

Round 2

Reviewer 2 Report

The authors have addressed most of the comments; they have also tried to make changes according to the reviewers’ suggestions. After revisions, the quality of the manuscript has been adequately enhanced. Therefore, the manuscript could be considered for publication in the Journal.

Author Response

Thank you for the recomendations.

Reviewer 3 Report

  1. Figures quality/resolution must be improved. Must satisfy journal standard. Figure captions must be elaborated with more experimental conditions.
  2. Authors must tabulate the performance of composite materials with literature. Must elaborate the table with more literature. 
  3. Reference style must be verified. 

Author Response

  1. Figures quality/resolution must be improved. Must satisfy journal standard. Figure captions must be elaborated with more experimental conditions.
  • The figure’s quality has been improved with the caption as suggested. 

  1. Authors must tabulate the performance of composite materials with literature. Must elaborate the table with more literature. 
  • The suggestion has been added in the manuscript with a highlight.

  1. Reference style must be verified.
  • The references styles have been modified